# Poverty, Material Hardship, and Children’s Outcomes: A Nuanced Understanding of Material Hardship in Childhood

**DOI:** 10.3390/children9070981

**Published:** 2022-06-30

**Authors:** Anika Schenck-Fontaine, Rebecca M. Ryan

**Affiliations:** 1Leibniz Institute for Educational Trajectories, 96047 Bamberg, Germany; 2Department of Psychology, Georgetown University, Washington, DC 20057, USA; rmr64@georgetown.edu

**Keywords:** material hardship, economic hardship, behavior problems, cognitive development, housing hardship, food insecurity, poverty

## Abstract

There are four distinct, related types of material hardship—basic expense hardship, food insecurity, housing hardship, and medical hardship. Extant research has not sufficiently accounted for the complex relationships between these different types of material hardship. Using 1997 and 2002 data from the Panel Study of Income Dynamics Child Development Study on a national sample of 3- to 17-year-old children (*N* = 3563), this study describes the prevalence of each type of material hardship, their relative correlations, and their associations with children’s behavior problems, and reading and math scores. Material hardship is more prevalent than income poverty and the four types of material hardship were only moderately correlated with each other. Only basic expense hardship, food security, and one type of medical hardship were associated with increased behavior problems. Only housing hardship was associated with lower math and reading scores. These findings highlight the need to more carefully investigate the distinctions between material hardship types in childhood and the importance of a diversified set of policy responses to protect children from the possible effects of distinct, but interrelated experiences of material hardship.

## 1. Introduction

Too many children in the United States grow up in families that are unable to provide for their basic needs, such as food, housing, and medical care. More than a third of children live in households that report at least one type of material hardship; nearly twice the number of children who live in households that are income poor [1]. The prevalence of material hardship among families with children is likely to be a topic of growing concern. Since the beginning of the coronavirus pandemic, levels of material hardship among U.S. families have increased [2,3], especially among already vulnerable low-wage workers [4]. Although material hardship is closely linked to income poverty, these are not synonymous. Some income poor families do not experience material hardship, while other families with incomes well above the poverty threshold struggle to meet their material needs [1,5,6,7]. Possible reasons for this distinction between income poverty and material hardship include differences in costs of living, employment transitions, unexpected economic shocks and unforeseen expenses, access to assets, and differences in family needs [8]

Scholars and policy makers in the United States commonly use income poverty, defined as household income below the federal poverty line (FPL) by household size, as a proxy for whether a family is unable to provide for its basic needs. Increasingly, scholars are also investigating material hardship as a more direct and relevant measure of children’s economic vulnerability than income poverty, and the evidence suggests that material hardship is harmful to children [9,10]. Measures of material hardship provide a direct and more relevant picture of a family’s ability to meet basic needs. The four major domains in which families can experience material hardship are basic expenses hardship, food insecurity, housing hardship, and medical hardship [11]. Although these four domains are correlated measures of economic vulnerability, they also reflect different causal processes and different economic experiences [11]. Research on the distinctions and potential interplay between these hardship types in children’s lives has been limited. Most studies have considered single hardship types in isolation or studied them jointly as a composite measure of material hardship, but a limitation of these approaches is that they do not provide insight into the distinct impacts that different types of hardship might have on children.

Comparing estimates of income poverty and material hardship shows that income poverty actually underestimates the share of children in the U.S. that are economically vulnerable. Using data from the Survey of Income and Program Participation (SIPP) from 1996 to 2011, Rodems and Shaefer report that 36% of children experienced some type of material hardship, while only 18% were income poor [1]. Material hardship was most prevalent among children in families with incomes above the poverty threshold. These findings are consistent with several studies that report that material hardship and income poverty are only modestly correlated [6,7,12]. Therefore, favoring income poverty over material hardship as a measure of economic vulnerability risks underreporting economic vulnerability.

### 1.1. The Effects of Material Hardship on Children

The four types of material hardship that are considered risk factors for children’s outcomes are basic expense hardship, food insecurity, housing hardship, and medical hardship. While these material hardships have been linked to worse academic, behavioral, and emotional outcomes for children e.g., [10,13,14,15], there is evidence that material hardship is more significantly associated with children’s behavioral outcomes than academic or cognitive outcomes [16].

Studies on the effects of material hardship on children have taken two primary approaches to estimating to what degree these hardships impact children. The composite or cumulative approach to measuring material hardship looks at the experience and effects of multiple types of material hardship without distinguishing between them. For example, studies using data from the Fragile Families and Child Wellbeing Study (FFCWS) show that, for socioeconomically vulnerable children and youth, any material hardship is associated with both internalizing and externalizing behavior problems [17,18], physical aggression [19], and anxiety and depression [20]. Studies using data from less disadvantaged samples of children also find that a composite measure of material hardship is associated with internalizing behavior problems [9,21], as well as social competence and behavior problems [7,16]. While most studies taking a composite approach have focused on associations with socioemotional outcomes, two studies report associations with grade retention [22] and lower executive function [21].

A second approach to material hardship research is to focus on the effects of a single type of material hardship in isolation of the others. A small number of studies has investigated basic expense hardship, which is commonly measured as the inability to meet basic needs, the inability to pay bills, and disconnected utilities [1,5,11]. Nearly a quarter of U.S. children are affected by basic expense hardship [1] and this type of hardship is associated with increased internalizing and externalizing behavior problems [10]. Basic expense hardship is also associated with a higher allostatic load in early childhood, which can present a long-term risk to cognitive skills, self-regulation, and mental health [23]. Basic expense hardship may affect children directly or indirectly. No studies have yet investigated possible mediating mechanisms, but research by Schenck-Fontaine and colleagues [10] supports the family stress and parenting behavior pathway.

Food insecurity is the type of material hardship that has been most widely studied, and it affects approximately 14% of U.S. children [1]. Food insecurity in the U.S. is typically measured using a version of the USDA Household Food Security Module, which captures both food insufficiency, as well as related worry [24]. In addition to associations with poor physical health e.g., [25,26], national surveys suggests that food insufficiency is associated with grade retention and worse math scores [27,28], worse social skills [28], and an increase in internalizing behavior problems [29]. Both poor nutrition and parents’ stress are probable causal pathways linking food insecurity to children’s cognitive and socioemotional outcomes [27,30].

Medical hardship, defined as lacking health insurance or the inability to access needed medical care [5,11], affects 14% of U.S. children [1]. Yet, medical hardship has not been widely studied with regard to children’s cognitive or socioemotional outcomes. There is some evidence that medical hardship places children at risk of a range of poor developmental outcomes [14]. Missed or delayed care is of particular concern because preventive medical care is a key mechanism for identifying and providing supports for possible developmental delays [31]. Additionally, qualitative research shows that lacking health insurance is a significant stressor for parents [14] and could, therefore, place children at risk of worse cognitive or socioemotional outcomes through family stress pathways.

Finally, housing hardship refers to a range of experiences that vary in severity, including unsafe or unsanitary housing conditions, crowded housing conditions, and eviction [5,11]. Housing hardship is the least prevalent hardship type among children, affecting only 8% of U.S. children [1]. Housing insecurity, measured as crowded housing and multiple moves, is associated with a greater risk for both cognitive and socioemotional developmental delays among very young children [32]. There is also evidence for associations between crowded housing and worse reading and math scores [33] and lower levels of completed schooling [34]. Poor quality housing has been linked to children’s psychological distress [35,36]. Housing quality may directly impact children’s health with consequences for their physical and socioemotional development, while other causal pathways, such as household chaos [35] or parenting behavior [37], have also been suggested to explain cognitive and socioemotional impacts.

### 1.2. Relationships among and Distinctions between Types of Material Hardship Types

Both of these common approaches to examining material hardship, using a composite measure or considering each hardship type in isolation, make the implicit assumption that all types of material hardship reflect a single underlying construct of insufficient resources to meet material needs. However, this is not supported by empirical evidence [11]. Using confirmatory factor analysis, Heflin and colleagues [11] demonstrate that basic expense hardship, food insecurity, housing hardship, and medical hardship are related, but distinct experiences of economic vulnerability emerge through different processes on different time horizons. The authors propose that basic expense hardship and food insecurity are both reflective of short-term economic shocks. Specifically, food insecurity reflects small, very short-term variations in resources and basic expense hardship is the result of compounding short-term shocks to resources. In contrast, housing hardship is the result of long-term economic constraints on decisions, and medical hardship is a combination of short-term economic shocks unrelated to short-term financial resources and longer-term structural and economic constraints. There is also evidence that the four material hardship types differ in their associations with children’s development. In addition to family stress, which is a likely mechanism that links all hardship types to children’s outcomes, food insecurity and housing hardship may also be associated with worse child outcomes through direct impacts on their health [25,27,35,37].

These distinctions between the types of hardship underscore the importance of examining their associations with children’s outcomes in relationship to each other. To our knowledge, only one study has investigated the associations between children’s outcomes and the four types of material hardship in relationship to each other. Using data from the FFCWS, Zilanwala and Pilkauskas [18] found that, taking other hardships into account, basic expense hardship and housing hardship were associated with children’s externalizing and internalizing behavior problems, but medical hardship and food insecurity were not. Though not focused on child outcomes, another study using the same approach with FFCWS data found that basic expense hardship was most prominently associated with parental depression [38].

### 1.3. Current Study

Using data from the 1997 and 2002 waves of the PSID-CDS, this study aims to describe the relationships between income, the four different types of material hardship, and their individual and relative associations with children’s behavior problems and reading and math scores. The results of this study contribute a more nuanced understanding of the diversity of material hardship experiences in childhood and its associations with children’s outcomes in a national sample. By distinguishing between poverty and near-poverty, this study also gives additional insight into the relationship between material hardship and income in childhood.

We first investigate the correlations between each type of material hardship and income poverty, which we expect to be only modest given extant evidence that income poverty and material hardship are distinct constructs [6,7,12]. Given the long-term nature of housing and medical hardship [11], we expect these to be least correlated with income poverty, while we expect the shorter-term hardships, basic expense hardship and food insecurity, to be more strongly correlated with income poverty. Next, we test the association between each type of material hardship and children’s outcomes, and net of income poverty. Given the direct effects of food insecurity and housing hardship on children’s health, we expect that only these are associated with children’s math and reading scores, but that all hardship types are associated with behavioral problems. Finally, we examine the association between each type of material hardship accounting for income poverty and all other types of hardship. Consistent with Zilanawala and Pilkauskas [18], we expect that only basic expense hardship is associated with children’s outcomes when accounting for other hardship types.

It is important to note that this study focuses on the effects of material hardship on children in the U.S. context, which differs in some significant ways from other wealthy countries, including European countries. Owing partly to weaker welfare and anti-poverty policies in the U.S., rates of childhood poverty and inequality are substantially higher in the U.S. relative to other wealthy countries [39,40,41]. This makes the U.S. an especially important context within which to study how material hardship is associated with children’s outcomes. There are also differences in the measurement of poverty and hardship across countries. Unlike in European countries, poverty in the U.S. is assessed using an absolute measure that does not account for changes in needs and costs of living that have occurred since the measure was set in 1960s [42]. As a result, the official poverty measure in the U.S. likely underestimates actual economic deprivation. The United States also does not have an established set of standard measures of material hardship, as is true in European countries, where there has been a tradition of distinguishing between income poverty and material hardship that began in the 1990s [43]. Because of these characteristics unique to the context of poverty and poverty measurement in the U.S., the findings of this study are specific to the U.S., but the theoretical implications can be applied to all national contexts.

## 2. Materials and Methods

### 2.1. Data

This study used data from the Panel Study of Income Dynamics Child Development Supplement (PSID-CDS). The PSID is a national longitudinal panel study that began in 1968 and was drawn from two independent, national probability samples that, when combined and weighted, can produce a nationally representative estimate for U.S. families in 1968 [44]. Since 1968, the PSID has followed these original respondents, as well as their children and, in some cases, grandchildren, which has resulted in a growth in the overall sample size over the years.

In 1997, the PSID supplemented its ongoing data collection with a longitudinal supplemental study, the PSID-CDS. In 1997, a family could provide information for up to two children in the household. The children and families were surveyed first in 1997 when most children were 0 to 12 years old. Additional CDS follow-up surveys were conducted in 2002 and 2007. Interviews and child assessments were completed in the home. More information about the CDS, as well as specific information about the reliability and validity of measures and scales used in the CDS, can be found in the PSID User’s Guides and technical documentation [44,45,46].

Data from the 1997 and 2002 data collection waves were used for the analyses in this study. In 1997, 2394 of the PSID families (88%) responded about their 3563 children. In 2002, 2019 of these families (91%) were surveyed about their 2907 children when most children were 5 to 17 years old. We restricted the analytical sample for both years to children who are between 3 and 17 years old. See Table 1 for descriptive statistics of the 1997 and 2002 samples. For each variable described below, the first choice was to use data collected in the same year as the child assessments (i.e., 1997 and 2002). For some variables, 1997 and 2002 data were not available. In these cases, we used data collected in the prior year which is noted below.

### 2.2. Child Cognitive Outcomes

Children’s reading and math skills were measured using the Woodcock–Johnson Psycho-Educational Battery-Revised (WJ-R) for all children aged six or older [46]. To measure reading skills, we used the constructed, age-standardized Broad Reading score. This score combines a child’s scores on the Letter-Word Identification subtest, which assess symbolic learning and reading identification, and the Passage Comprehension subtest, which measures reading comprehension and vocabulary. To measure math skills, we used the age-standardized Applied Problems score, which assesses math reasoning, achievement, and knowledge. All scores are standardized to have a mean of 100 and a standard deviation of 15. For the estimation models, we convert all standardized scores to z-scores with a mean of 0 and a standard deviation of 1 to allow for comparisons with the behavioral outcomes described below.

### 2.3. Child Behavioral Outcomes

Children’s behavior problems were measured using the Behavior Problems Index (BPI) for all children aged three or older. This parent- or caregiver-reported measure asks 30 questions about children’s behavior, including anxiousness, aggression, and obsessiveness. The questions and response categories are: (1) often true, (2) sometimes true, and (3) not true. Based on confirmatory factor analysis results, 27 of the 30 items are used to construct two subscales: internalizing or withdrawn behaviors, and externalizing or aggressive behaviors. To allow for comparisons across coefficients, we converted the raw scores to z-scores with a mean of 0 and a standard deviation of 1 for the estimation models.

### 2.4. Income Poverty

To measure a household’s poverty status, we used information from the main PSID survey on total family income, including taxable, transfer, and social security income of all family members. Because data on family income was not available for 1997, we used income information from 1996. Using the reported total family income, we coded a family’s poverty status relative to the federal poverty line for that year (i.e., the 1996 or 2002 federal poverty line). We define poor families as those families with total family income below 100% of the federal poverty line (FPL). Near-poor families are those with total family income between 100% and 250% of the FPL. Not poor families are those with total family income at 250% of the FPL or above.

### 2.5. Material Hardship

We include two dichotomous measures of basic expense hardship using information available in 1997 and 2002 data. Respondents were asked whether the family borrowed money from friends or relatives in the prior year (0 = no, 1 = yes). Respondents were also asked how much money they have left to make ends meet at the end of a month (1 = some money left over, 2 = just enough to make ends meet, 3 = not enough to make ends meet). Using this information, we constructed a dichotomous indicator for not being able to make ends meet (0 = no, 1 = yes). Food insecurity is measured using 1997 and 2001 data from the 18-item USDA Household Food Security Module, which categorizes households as having high food security (score of 0), marginal food security (score of 1 to 2), low food security (score 3 to 7), and very low food security (score 8 to 18). While marginal food security typically reflects worry about food sufficiency, low food security reflects some reduced quality or variety of food, and very low food security reflects reduced quality, as well as reduced intake of food [47]. For the purpose of this study, we focus on food insufficiency and consider all households experiencing low or very low food security as food insecure (0 = no, 1 = yes).

Housing hardship was measured using two items. First, we constructed a dichotomous indicator for crowded housing using data from 1996 and 2002 and the standard definition of more than one person per room, excluding kitchens and bathrooms (0 = no, 1 = yes) [48]. Second, families were asked whether they moved in with someone else because of economic problems (0 = no, 1 = yes). Finally, medical hardship was also measured with two items based on data from 1997 and 2002. We include a dichotomous indicator for whether all members of the family lacked health insurance (0 = no, 1 = yes) in the two years prior to data collection, as well as an indicator of whether the family postponed medical care in the prior year due to economic problems (0 = no, 1 = yes). Because health hardship is measured at the family-level, unlike other domains of hardship, this may reflect both direct and indirect effects on the child.

### 2.6. Control Variables

We included a set of control variables in all estimation models. We included information about the parent’s or caregiver’s age, their level of education, and their marital status. Child characteristic controls included the child’s age, sex, and race and ethnicity. Finally, household structure controls included the total number of children and adults living in the household.

### 2.7. Analytical Strategy

First, we conducted correlation analyses (available in Appendix A) to explore the relationships between dichotomous indicators for different types of material hardship and income poverty using a phi coefficient [49]. Second, we examined the prevalence of each type of material hardship among poor, near-poor, and not poor families. Third, we ran regression models estimating the association between material hardship types, poverty status, and children’s cognitive and behavioral outcomes. To do this, we first estimated the association between the indicators of each type of material hardship and child outcomes net of income poverty status and all control variables (Models 1–4 in each table). Finally, we estimated models that include all indicators of material hardship and income poverty simultaneously (Model 5 in each table). Separate models were estimated for 1997 and 2002 data.

All child outcomes are continuous, and we used linear multivariate regression for all estimation models. Standard errors are clustered at the household level to account for children in the sample who are living in the same household, and analyses are weighted using PSID-provided weights to adjust for differential sampling and attrition probabilities. For most variables, less than 10% of data were missing. However, for several variables, including not having enough money to make ends meet, borrowing money from friends, and postponing medical care, up to 29% of data were missing. While we could not assume that the data are missing completely at random, we chose to use listwise deletion in our preferred specifications. In an alternative specification, we employed a Full Information Maximum Likelihood (FIML) approach, which uses all observed information to draw inferences about the missing responses [50]. The FIML-estimated models produced results that are highly similar to those produced by models estimated using listwise deletion, which suggests that listwise deletion does not produce substantially more biased estimates when the data are missing for predictor variables [51].

## 3. Results

Table 1 presents weighted summary statistics for all analysis variables for the 1997 and 2002 samples. Borrowing money, an indicator of basic expense hardship, is the most prevalent of all material hardships (30.2% in 1997 and 20.2% in 2002). This is substantially higher than the share of children living in income poor households (21.1% and 15.3% in 1997 and 2002, respectively). Most other types of material hardship are less prevalent than income poverty in both years. There are some differences between the 1997 and 2002 analytic samples. As a matter of design, the children are older in 2002 (6.8 vs. 12.3 years old in 1997 and 2002, respectively). The average income is higher in 2002 than in 1997 (USD 73,126.32 compared to USD 50,937.45), though this is at least partly due to inflation. Overall, the prevalence of most types of material hardship is somewhat lower in 2002 than in 1997, with the exception of postponing medical care.

The weighted phi correlation coefficients for income poverty and each indicator of material hardship for 1997 and 2002 are shown in Appendix A Table A1. Overall, the correlations between income poverty and each type of material hardship, as well as between each type of material hardship, are weak to moderate. However, basic expense hardship indicators were correlated with food insecurity and one indicator of medical hardship, postponing medical care due to economic problems.

### 3.1. Prevalence of Material Hardship by Poverty Status

Table 2 shows that poor families (i.e., with incomes below 100% of the FPL) are not the only ones to experience material hardship. Poor families are more likely to experience material hardship than near-poor and not poor families, but near-poor families (i.e., with incomes between 100% and 250% of the FPL) nevertheless experience notably higher rates of all types of material hardship than not poor families. In 1997, nearly 50% of poor families borrowed money and 28% had no money left at the end of a month to make ends meet. Near-poor families similarly experienced very high levels of basic expense hardship, with 43% borrowing money and 21% not having money left at the end of a month. This is in contrast with not poor families, among whom 15% borrowed money and only 6% had no money left at the end of a month. While the prevalence of basic expense hardship was lower for all families in 2002, this was especially true for near-poor families. The prevalence of food insecurity for the near-poor falls in between that among poor and not poor families (30% among poor families, 13% among near-poor families, and 3% among not poor families in 1997). Rates of food insecurity were similar in 2002, albeit somewhat higher for near-poor families.

Housing hardship was a relatively rare experience among near-poor families in 1997. In 2002, the prevalence of housing hardship increased for all families and especially so for near-poor families. For example, 15% of poor families lived in crowded housing in 1997, compared to only 6% of near-poor and 2% of poor families. In 2002, 33% of poor families lived in crowded housing, compared to 17% of near-poor families and just 3% of not poor families. The patterns are similar for those having moved in with others due to economic problems, though the prevalence is overall much lower than for crowded housing. Finally, medical hardship was as common or more common for near-poor families as poor families in 1997, in which 7% of poor families had no health insurance compared to 6% of near-poor families, and 14% of poor families delayed medical care compared to 18% of near-poor families. This changed somewhat in 2002, when a greater share of poor families had no health insurance than near-poor families (13% vs. 6%), but still more near-poor families delayed medical care than poor families (20% vs. 14%).

### 3.2. Associations between Income Poverty, Material Hardship Types, and Child Outcomes

#### 3.2.1. Externalizing Behavior Problems

Table 3 presents the results for the regressions models predicting children’s externalizing behavior problems. Panel A presents the regressions results for 1997. Model 1 shows that borrowing money was associated with 0.26 standard deviation in higher levels of externalizing behavior problems (*p* < 0.01), and not having enough money at the end of a month was associated with 0.29 standard deviation in higher levels (*p* < 0.01). Model 2 shows a similar pattern for food insecurity, which was associated with 0.28 standard deviation higher externalizing behavior problems (*p* < 0.01). Model 3 results show that living in crowded housing was associated with 0.37 standard deviation in higher levels of externalizing behavior problems (*p* < 0.05). There was no significant association between medical hardship and externalizing behavior problems (Model 4). Finally, Model 5 includes all indicators of material hardship. The results of this model show that borrowing money (b = 0.25, *p* < 0.01) and not having enough money at the end of a month (b = 0.26, *p* < 0.05) were significantly associated with higher levels of externalizing behavior problems, holding income poverty status and all other types of material hardship constant.

Panel B of Table 3 presents the regression results for 2002. Model 1 shows that borrowing money was associated with 0.34 standard deviation in higher levels of externalizing behavior problems (*p* < 0.01). Food insecurity and housing hardship were not associated with externalizing behavior problems over and above income poverty in 2002 (Models 2 and 3). However, postponing medical care was marginally associated with significantly higher levels of externalizing behavior problems (b = 0.24, *p* < 0.10) (Model 4). Finally, Model 5 results show that only borrowing money was associated with higher levels of externalizing behavior problems (b = 0.32, *p* < 0.1) when accounting for income poverty and all other types of material hardship.

#### 3.2.2. Internalizing Behavior Problems

Panel A of Table 4 presents the results for the regressions models predicting children’s internalizing behavior problems in 1997. Model 1 shows that borrowing money was associated with 0.25 standard deviation in higher levels of internalizing behavior problems (*p* < 0.01) and not having money left at the end of a month was associated with 0.30 standard deviation in higher levels of internalizing behavior problems (*p* < 0.01). Food insecurity and housing hardship were not significantly associated with internalizing behavior problems (Models 2 and 3). Only one indicator of medical hardship (Model 4), postponing medical care, was significantly associated with internalizing behavior problems (b = 0.19, *p* < 0.1). Finally, Model 5 includes all indicators of material hardship and shows that only basic expense hardship was significantly associated with higher levels of internalizing behavior problems net of income poverty. Borrowing money was associated with 0.24 standard deviation in higher levels of internalizing behavior problems (*p* < 0.01) and not having money left at the end of a month was associated with 0.28 standard deviation in higher levels of internalizing behavior problems (*p* < 0.05).

Table 4, Panel B shows the regression results for internalizing behavior problems in 2002. Borrowing money was associated with 0.46 standard deviation in higher levels of internalizing behavior problems (*p* < 0.01), while not having money left at the end of a month was associated with 0.25 standard deviation in higher levels of internalizing behavior problems (*p* < 0.1) (Model 1). Food insecurity (b = 0.32, *p* < 0.5), moving in with others (b = 0.66, *p* < 0.05), and postponing medical expenses (b = 0.55, *p* < 0.01), 0 were each significantly associated with higher levels of internalizing behavior problems net of income poverty. However, Model 5 shows that when all hardship types are accounted for, only borrowing money (b = 0.38, *p* < 0.01) and postponing medical expenses (b = 0.36, *p* < 0.05) were significantly associated with internalizing behavior problems.

#### 3.2.3. Reading Scores

Table 5 presents the regression results for broad reading scores. Panel A shows that, in 1997, there were no significant associations between material hardship indicators and reading scores. In 2002 (Panel B), income poverty was marginally, but consistently associated with reading scores. However, only having moved in with others was significantly associated with 0.90 standard deviation in lower reading scores (*p* < 0.01) (Model 4). This marginally significant association remained even when all other material hardship indicators are controlled for in Model 5 (b = −0.86, *p* < 0.05).

#### 3.2.4. Applied Problems Scores

Table 6 presents the regression results for applied problems scores. Panel A shows that, similar to reading scores, no material hardship indicators were significantly associated with applied problems scores in 1997. Panel B shows that, in 2002, being poor was consistently, but marginally associated with lower applied problem scores. Model 3 shows that having moved in with others was significantly associated with 0.53 standard deviation in lower applied problems scores (*p* < 0.05). Model 5, which includes poverty and all indicators of material hardship, shows that moving in with others (b = −0.50, *p* < 0.1) was associated with lower applied problem scores. On the other hand, not having health insurance was associated with 0.31 standard deviation in higher applied problems scores (*p* < 0.05).

### 3.3. Specification Checks

A major difference between the 1997 and 2002 samples is the age of the children in the sample. On average, children were 5.5 years older in 2002 than in 1997. To assess whether this age difference explains some of the differences in the associations between income poverty, material hardship types, and children’s outcomes, we re-estimate all models for internalizing and externalizing behavior problems and reading scores for both years using a restricted sample of only 6- to 12-year-olds (results available upon request). In our preferred specification that includes all 3- to 17-year-old children and youths, we observed that postponing medical care due to economic problems is significantly associated with more internalizing and externalizing behavior problems in 2002, but not in 1997 (see Table 4 and Table 5). We found the same pattern when we compared only 6- to 12-year-olds between 1997 and 2002, which suggests that this difference between the years cannot be explained by the different ages of the samples. We had also observed in our preferred specification that food insecurity is associated with internalizing behavior problems and crowded housing is associated with reading scores in 2002, but not in 1997. Neither of these differences between the years remained when we restricted both samples to 6- to 12-year-olds, suggesting that these differences between the years likely reflect age-specific associations.

We also conducted a series of additional specification checks (results are available upon request). We constructed a range of alternative specifications for the poverty categories, including a four-category variable that distinguishes between deep-poor families with incomes below 50% of the FPL and poor families with incomes between 50% and 100% of the FPL. The results of these models generally show that children in deep poverty have greater levels of material hardship and present worse behavior problems and have lower reading and math scores than poor and near-poor children. While these models provide a more nuanced understanding of the associations between income and child outcomes, these models are also underpowered and not central to the research questions of this study. Therefore, we prefer the specifications that do not distinguish between deep poverty and poverty. Additionally, we estimated the models using different near-poor definitions, including 100% to 200% of FPL and 130–200% FPL. While the results differed somewhat between specifications, they are substantively consistent with the results presented here.

## 4. Discussion

This study describes the prevalence of four different types of material hardships (i.e., basic expense hardship, food insecurity, housing hardship, and medical hardship), their correlations with each other, and their associations with children’s cognitive and behavioral outcomes in a national sample of children in the United States. Our results support the growing consensus that income poverty and material hardship do not necessarily reflect the same experience of economic vulnerability [1,5,6,11]. Moreover, our results support the theoretical model proposed by Heflin and colleagues [11] that describes the four types of material hardship as different, but related, aspects of economic vulnerability that emerge from distinct social processes. While we found overall low to moderate correlations between the different material hardship types, the magnitude of correlations varied between types. Specifically, we found stronger correlations among basic expense hardship, food insecurity, and postponing medical care due to economic problems, consistent with Heflin and colleagues’ [11] argument that these types of hardship reflect relatively short-term economic shocks. Basic expense hardship reflects short-term shocks, whereas housing hardships likely emerge due to longer-term economic constraints [11].

Our correlation results reinforce the argument that different types of material hardship reflect different aspects of economic vulnerability that may each play a different role in children’s lives and that more care is needed in the measurement of material hardship in child development research. We build on Heflin et al.’s framework [11] by showing that even the correlations between indicators of the same type of material hardship are moderate at best. These modest correlations suggest that, as is the case with the two types of medical hardship, different processes may explain different indicators and, thus, we may expect different effects on children. Therefore, scholars must be careful not to conflate distinct hardship experiences, as has been the trend in research on material hardship in childhood, to provide nuanced evidence that can inform effective interventions and policies. Despite important distinctions between types of material hardship, these are correlated, and together form the landscape of economic vulnerability in childhood. Examining them in isolation of each other could lead to misleading interpretations.

The regression results show that only basic expense hardship, and borrowing money in particular, is consistently associated with externalizing and internalizing behavior problems when accounting for income poverty and other types of material hardship. Postponed medical care, which reflects short-term economic constraints similar to basic expense hardship [11] is also associated with internalizing behavior problems, but this was significant only in 2002. These results are only partly consistent with the findings by Zilanwala and Pilkauskas [18], who reported significant associations between basic expense hardship and housing hardship with behavior problems among economically vulnerable families.

We also found that income poverty and one indicator of housing hardship are negatively associated with reading and applied problems scores. Though we predicted a significant association between food insecurity and these cognitive outcomes, we found no consistent significant associations. Consistent with the prior literature on housing hardship [33,34], moving in with others, an indicator of significant housing hardship, is strongly associated with worse reading and applied problems scores even when all other material hardship types were controlled for. We also found a positive association between not having medical insurance and applied problems scores, but we believe this association to be likely spurious.

Our results add important nuance to a previous finding that material hardship influences children’s socioemotional outcomes, while income poverty influences children’s cognitive outcomes [16]. Instead, it is important not to conflate different types of material hardship experiences. More research is necessary into the possible explanations some material hardships are associated with behavior problems, but our results provide a potential clue. Both material hardship types that we found to be associated with children’s behavior problems reflect short time horizons, whereas housing hardship, which reflects a longer time horizon, is the only material hardship type that we found to be associated with cognitive outcomes. One possible explanation is that short-term economic shocks lead to more significant stress for a family than longer-term material hardships for which families may have had time to put coping strategies in place. Family stress is, in turn, most significantly associated with children’s behavioral problems [16]. However, there is also evidence that families are more resilient to economic shocks in the short-term [52] and additional research is needed to understand why short-term shocks appear more strongly associated with children’s behavior problems.

On the other hand, housing hardship, which reflects longer-term economic processes [11] may influence the resources parents are able to invest in their children’s learning and development of reading and math skills. Housing hardship is also associated with a lack of access to high quality schools, unscheduled school transfers, and truancy, all of which have consequences for children’s learning over a longer time horizon [53,54].

We noted some differences in the associations between material hardship and children’s outcomes between 1997 and 2002. For example, food insecurity and housing hardship were associated with externalizing behavior problems only in 1997 and internalizing only in 2002. Not having enough money left at the end of a month to make ends meet was only associated with behavior problems in 1997. Another notable difference between 1997 and 2002 was that moving in with others was only significantly associated with reading and applied problems scores in 2002. These differences between the years may reflect the differential importance of material hardships at different ages or developmental stages. On average, children were six years older in 2002. It is possible that school-aged children are more sensitive to housing hardship, whereas young children are more sensitive to food insecurity or some types of basic expense hardship. On the other hand, these differences may also reflect differences in the policy and economic context between 1997 and 2002. For example, 2002 data were collected during the tail end of an economic recession [55], during which families may have postponed large expenses, including those related to medical care. While the Children’s Health Insurance Program (CHIP), created in 1997, should have buffered this effect somewhat, it is possible that the additional economic strain introduced by the economic recession led to greater stress associated with postponing medical expenses and, in turn, greater association with children’s behavior problems in 2002. Additional analyses are needed to explore both age differences in the associations between different types of material hardship and children’s outcomes and differences across economic and policy contexts. Further research also needs to be conducted to examine the associations between different types of material hardship and children’s outcomes among children with special health care needs and other populations vulnerable to higher levels of hardship.

This study is not without limitations. First, the PSID sample is somewhat more white and economically advantaged relative to current national norms, which likely means that the associations we find are subject to a downward bias and cannot be directly generalized to today’s population. Second, because of the inclusion of multiple correlated predictors in a single model, the analyses may be statistically underpowered and, as a result, the estimates may be imprecise. As such, it is possible that we underestimate the significance of the associations between poverty, material hardship, and children’s outcomes. To account for this, we report marginally significant estimates (i.e., *p* < 0.1), which may be practically meaningful. Third, additional research is necessary to assess whether the associations we find are relevant for children’s long-term outcomes.

Fourth, the 1997 and 2002 PSID-CDS data are nearly 25 years old and do not reflect the current economic or policy context, which limits our ability to generalize from these findings. These waves of PSID-CDS data provide the most recent national U.S. data source of which we are aware that includes information on all types of material hardship and measures of both children’s socioemotional and cognitive outcomes. That the prevalence estimates we report are consistent with more recent data [1] strengthens our confidence in the relevance of these results in today’s context. Nevertheless, it is important that these analyses of the associations between material hardship domains and children’s outcomes be replicated using recent national data that capture the changes introduced by the recent significant recessions and the pandemic. To ensure that U.S. research about material hardship in childhood is relevant and valid, investment in new national data sources that include information about material hardship and child outcomes is critical.

Fifth, the descriptive analyses presented here do not take advantage of the longitudinal nature of the PSID-CDS to rule out potential confounders in the relationship between material hardship types and children’s outcomes. The choice to focus on descriptive analyses was intentional. While much of the existing material hardship literature has focused on establishing plausibly causal links between material hardship and children’s outcomes, e.g., [9,10,17,18], there has not been enough research that carefully describes the experience of material hardship in childhood. A notable exception to this is a recent study that describes the relative prevalence of each type of material hardship among children in the U.S. [1], though this study did not examine their relative correlations or associations with children’s outcomes. Descriptive research that carefully examines the nuanced distinctions between material hardship types is not only instructive for further research, but it is also essential for guiding the design and targeting of policies intended to protect economically vulnerable children.

Finally, the measures of material hardship used in this study particularly reflect the context of poverty and the available measures of material hardship in the U.S. Medical hardship, for example, is unique to the United States, where, despite public insurance programs for low-income individuals, many are nevertheless unable to afford adequate health care [56,57]. As such, the particular degree of overlap between the domains of material hardship and their associations with child outcomes are unique to the U.S. context. However, the theoretical implication that different domains of hardship may be differentially associated with children’s outcomes and should not be conflated can be applied to all economic and policy contexts. Further research using data from other national contexts is needed to investigate similarities and differences in the relationships between domains of material hardship and child outcomes across contexts.

This study makes several important contributions to the literature on material hardship in childhood. By using data from a national study of families with children, we add to the body of evidence that many children experience material hardship although their families would not traditionally be considered economically vulnerable because their incomes are well above the FPL. We add nuance to this literature by showing that the four major types of material hardship—basic expense hardship, food insecurity, housing hardship, and medical hardship—are distinct and only moderately correlated. We show that material hardships that more likely reflect short-term economic shocks are associated with children’s behavior problems, while housing hardship, which reflects longer-term economic constraints, is associated with children’s reading and math scores. These findings underscore the importance of distinguishing between material hardship types, as conflating these could lead to misleading interpretations. From a research perspective, these findings highlight the need to more carefully investigate both the etiology of each material hardship type and the mechanisms linking different types of material hardship to children’s outcomes. From a policy perspective, that different types of material hardship likely emerge from different economic and structural processes and, in turn, are also differentially associated with children’s outcomes necessitates a diverse basket of both cash and in-kind supports to protect economically vulnerable children. For example, while programs to buffer short-term cash flow problems, such as the short-lived Advance Child Tax Credit, may best protect children from basic expense hardship and food insecurity, we argue that affordable housing policies and universal health care remain necessary goals to protect children from longer-term material hardships.

## Figures and Tables

**Table 1 children-09-00981-t001:** Sample descriptive statistics, 1997 and 2002.

	1997		2002	
	Mean	SD	Mean	SD
Child Characteristics				
Age	6.79	3.77	12.32	3.72
Male (%)	50.31	50.00	49.71	50.01
Non-Hispanic White (%)	63.43	48.17	62.48	48.44
Black (%)	16.07	36.73	19.43	39.58
Other Race (%)	7.47	26.29	7.46	26.29
Hispanic (%)	13.04	33.67	10.62	30.84
Household and PCG Characteristics				
PCG—Male (%)	2.76	16.38	5.63	23.06
PCG—Age	34.15	7.26	38.44	7.43
PCG—Education (years)	12.81	2.70	12.98	2.73
HH Head married (%)	72.09	44.86	74.17	43.78
HH size	4.33	1.28	4.40	1.34
Economic Circumstances				
Crowded housing (%) *	11.45	31.85	9.94	29.92
Moved in with others (%)	3.20	2.64	2.61	15.93
Borrowed money (%)	30.20	45.93	20.22	40.17
No money at end of month (%)	12.83	33.45	12.55	33.14
No health insurance (%)	4.68	21.12	4.53	20.80
Postponed medical care (%)	8.79	28.32	10.33	30.44
Food insecure (%) *	11.86	32.33	10.96	31.25
Household income *	$50,937.45	$52,675.94	$73,126.32	$92,002.61
Child Outcomes				
BPI—Internalizing Problems	2.58	2.64	3.32	3.24
BPI—Externalizing Problems	5.62	3.83	5.66	4.16
Broad Reading Score (Std.)	106.31	18.06	105.90	17.61
Applied Problems Score (Std.)	107.07	17.77	104.66	17.31
	*N* = 1520–2732		*N* = 2537–2893	

Notes: Weighted averages. PCG: Primary Care Given, HH: household, BPI: Behavior Problem Index. * Measurement in prior year.

**Table 2 children-09-00981-t002:** Prevalence of material hardship by poverty status.

	Poor		Near-Poor		Not Poor	
	Mean	SD	Mean	SD	Mean	SD
Panel A: 1997						
Borrowed money (%) *	48.79	50.06	42.76	49.53	15.41	36.12
No money at end of month (%)	27.59	44.77	20.51	40.42	5.57	22.95
Food insecure (%)	29.78	45.78	13.39	34.08	2.84	16.61
Crowded housing (%) *	14.76	35.51	5.88	23.54	2.20	14.67
Moved in with others (%)	5.32	22.48	3.13	17.45	0.83	9.09
No health insurance (%)	6.95	25.45	5.55	22.92	1.15	10.66
Postponed medical care (%)	13.93	34.68	18.10	38.55	6.36	24.43
	*N* = 318–515		*N* = 457–762		*N* = 747–1182	
Panel B: 2002						
Borrowed money (%)	41.26	49.33	33.10	47.10	13.58	34.28
No money at end of month (%)	31.18	47.19	16.36	37.03	9.71	23.14
Food insecure (%) *	31.41	46.51	17.81	38.30	3.78	19.08
Crowded housing (%)	33.23	47.19	17.03	37.62	2.75	16.37
Moved in with others (%)	6.72	25.09	5.30	22.42	0.90	9.44
No health insurance (%)	12.55	33.19	5.63	23.07	1.05	10.18
Postponed medical care (%)	13.47	34.21	19.93	39.99	4.38	20.49
	*N* = 228–262		*N* = 452–533		*N* = 712–825	

Notes: Weighted averages. * Measurement in prior year. Income poor families have incomes less than 100% FPL; near-poor families have incomes between 100% and 250% of FPL; not poor families have incomes above 250% FPL.

**Table 3 children-09-00981-t003:** Associations between material hardship, income poverty, and externalizing behavior problems.

	Panel A: 1997						Panel B: 2002					
	(1)	(2)	(3)	(4)	(5)	(6)	(1)	(2)	(3)	(4)	(5)	(6)
	Income Only	Basic Expense	Food Insecurity	Housing	Medical	All Hardship	Income Only	Basic Expense	Food Insecurity	Housing	Medical	All Hardship
Poor	0.154	0.075	0.097	0.160	0.167	0.087	0.471 **	0.370 *	0.441 **	0.459 **	0.437 **	0.366 *
	(0.112)	(0.124)	(0.114)	(0.129)	(0.135)	(0.131)	(0.201)	(0.209)	(0.198)	(0.199)	(0.205)	(0.203)
Near-poor	0.066	0.069	0.040	0.134	0.155	0.109	0.063	0.025	0.049	0.048	0.013	0.019
	(0.078)	(0.093)	(0.078)	(0.096)	(0.099)	(0.098)	(0.125)	(0.122)	(0.130)	(0.124)	(0.127)	(0.124)
Borrowed money		0.259 ***				0.249 ***		0.342 ***				0.316 **
		(0.085)				(0.089)		(0.132)				(0.136)
Not enough money		0.290 ***				0.261**		0.036				0.001
		(0.106)				(0.112)		(0.137)				(0.146)
Food insecure			0.280 ***			−0.036			0.139			0.014
			(0.093)			(0.117)			(0.154)			(0.171)
Crowded housing				0.374 **		0.226				−0.022		0.077
				(0.159)		(0.171)				(0.233)		(0.221)
Moved in with others				0.038		−0.084				0.384		0.220
				(0.170)		(0.186)				(0.289)		(0.244)
No health insurance					0.020	0.057					−0.106	−0.121
					(0.150)	(0.154)					(0.221)	(0.202)
Postponed medical expense					0.115	0.009					0.236 *	0.116
					(0.102)	(0.097)					(0.137)	(0.139)

Notes: Weighted estimates. All coefficients in standard deviation units. Clustered standard errors in parentheses. *** *p* < 0.01, ** *p* < 0.05, * *p* < 0.1.

**Table 4 children-09-00981-t004:** Associations between material hardship, income poverty, and internalizing behavior problems.

	Panel A: 1997						Panel B: 2002					
	(1)	(2)	(3)	(4)	(5)	(6)	(1)	(2)	(3)	(4)	(5)	(6)
	Income only	Basic Expense	Food Insecurity	Housing	Medical	All Hardship	Income only	Basic Expense	Food Insecurity	Housing	Medical	All Hardship
Poor	0.261 **	0.122	0.237 *	0.215 *	0.191	0.120	0.603 ***	0.429 **	0.533 ***	0.589 ***	0.540 ***	0.412 **
	(0.128)	(0.113)	(0.129)	(0.117)	(0.124)	(0.121)	(0.193)	(0.184)	(0.196)	(0.194)	(0.192)	(0.186)
Near-poor	0.185 **	0.186 **	0.163 **	0.261 ***	0.271 ***	0.220 **	−0.011	−0.105	−0.046	−0.025	−0.121	−0.156
	(0.074)	(0.087)	(0.074)	(0.089)	(0.090)	(0.089)	(0.125)	(0.122)	(0.128)	(0.123)	(0.124)	(0.121)
Borrowed money		0.254 ***				0.235 ***		0.460 ***				0.379 ***
		(0.078)				(0.079)		−0.131				(0.126)
Not enough money		0.297 ***				0.284**		0.253*				0.155
		(0.110)				(0.116)		−0.143				(0.158)
Food insecure			0.180			−0.074			0.323 **			0.128
			(0.115)			(0.124)			(0.163)			(0.174)
Crowded housing				0.273		0.179				−0.134		−0.108
				(0.188)		(0.196)				(0.236)		(0.212)
Moved in with others				−0.137		−0.265				0.662 **		0.414
				(0.147)		(0.166)				(0.325)		(0.309)
No health insurance					−0.021	0.027					−0.154	−0.122
					(0.144)	(0.158)					(0.259)	(0.238)
Postponed medical expense					0.188 *	0.099					0.551 ***	0.360 **
					(0.100)	(0.098)					(0.156)	(0.159)

Notes: Weighted estimates. All coefficients in standard deviation units. Clustered standard errors in parentheses. *** *p* < 0.01, ** *p* < 0.05, * *p* < 0.1.

**Table 5 children-09-00981-t005:** Association between material hardship, income poverty, and reading scores.

	Panel A: 1997						Panel B: 2002					
	(1)	(2)	(3)	(4)	(5)	(6)	(1)	(2)	(3)	(4)	(5)	(6)
	Income only	Basic Expense	Food Insecurity	Housing	Medical	All Hardship	Income only	Basic Expense	Food Insecurity	Housing	Medical	All Hardship
Poor	−0.117	−0.161	−0.140	−0.152	−0.163	−0.166	−0.231 *	−0.233 *	−0.231 *	−0.233 *	−0.217 *	−0.236 *
	(0.137)	(0.149)	(0.135)	(0.151)	(0.147)	(0.149)	(0.124)	(0.127)	(0.126)	(0.126)	(0.125)	(0.129)
Near-poor	−0.171 *	−0.163	−0.164 *	−0.158	−0.160	−0.146	−0.053	−0.091	−0.053	−0.058	−0.043	−0.106
	(0.095)	(0.119)	(0.095)	(0.109)	(0.114)	(0.124)	(0.140)	(0.134)	(0.143)	(0.134)	(0.135)	(0.129)
Borrowed money		−0.038				−0.069		−0.073				−0.021
		(0.109)				(0.109)		(0.119)				(0.125)
Not enough money		−0.018				−0.022		0.096				0.120
		(0.126)				(0.129)		(0.159)				(0.150)
Food insecure			−0.100			0.073			0.001			0.053
			−0.039			(0.126)			(0.125)			(0.140)
Crowded housing			(0.113)	−0.089		−0.077				0.211		0.095
				(0.213)		(0.207)				(0.216)		(0.210)
Moved in with others				−0.566		−0.647				−0.904 ***		−0.864 **
				(0.428)		(0.419)				(0.325)		(0.357)
No health insurance					0.143	0.187					0.164	0.245
					(0.184)	(0.198)					(0.227)	(0.192)
Postponed medical expense					−0.131	−0.070					−0.171	−0.135
					(0.137)	(0.133)					(0.164)	(0.142)

Notes: Weighted estimates. All coefficients in standard deviation units. Clustered standard errors in parentheses. *** *p* < 0.01, ** *p* < 0.05, * *p* < 0.1.

**Table 6 children-09-00981-t006:** Association between material hardship, income poverty, and applied problems scores.

	Panel A: 1997						Panel B: 2002					
	(1)	(2)	(3)	(4)	(5)	(6)	(1)	(2)	(3)	(4)	(5)	(6)
	Income only	Basic Expense	Food Insecurity	Housing	Medical	All Hardship	Income only	Basic Expense	Food Insecurity	Housing	Medical	All Hardship
Poor	−0.232 ***	−0.168 *	−0.231 ***	−0.185 *	−0.196 **	−0.182 *	−0.247 *	−0.204	−0.262 **	−0.244 *	−0.236 *	−0.238 *
	(0.088)	(0.101)	(0.089)	(0.098)	(0.096)	(0.100)	(0.127)	(0.131)	(0.128)	(0.127)	(0.127)	(0.130)
Near-poor	−0.291 ***	−0.260 ***	−0.288 ***	−0.285 ***	−0.312 ***	−0.288 ***	−0.182	−0.161	−0.189	−0.182	−0.194	−0.211 *
	(0.071)	(0.088)	(0.072)	(0.085)	(0.088)	(0.091)	(0.118)	(0.118)	(0.122)	(0.115)	(0.118)	(0.116)
Borrowed money		−0.041				−0.036		−0.180				−0.187
		(0.067)				(0.070)		(0.124)				(0.116)
Not enough money		−0.111				−0.109		0.010				0.032
		(0.086)				(0.090)		(0.121)				(0.124)
Food insecure			−0.038			0.045			0.072			0.176
			(0.101)			(0.137)			(0.122)			(0.123)
Crowded housing				−0.173		−0.113				0.137		0.085
				(0.113)		(0.109)				(0.186)		(0.177)
Moved in with others				−0.064		−0.084				−0.553 **		−0.501 *
				(0.199)		(0.198)				(0.273)		(0.271)
No health insurance					0.007	−0.009					0.279	0.306 **
					(0.103)	(0.109)					(0.187)	(0.152)
Postponed medical expense					−0.031	0.0120					−0.194	−0.145
					(0.091)	(0.089)					(0.131)	(0.132)

Notes: Weighted estimates. All coefficients in standard deviation units. Clustered standard errors in parentheses. *** *p* < 0.01, ** *p* < 0.05, * *p* < 0.1.

## Data Availability

The data presented in this study are openly available in the Inter-university Consortium for Political and Social Research at https://doi.org/10.3886/ICPSR37142.v1; accessed on (10 December 2020).

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
