# Peer review of "Poverty, Material Hardship, and Children’s Outcomes: A Nuanced Understanding of Material Hardship in Childhood"

_children, 2022, doi:10.3390/children9070981_

Round 1

Reviewer 1 Report

Excellent manuscript on a very important topic. I have only one suggestion: in discussing the significant effect of housing hardship on cognitive outcomes (lines 488-502), I would strongly recommend considering access to schools / changing schools (due to housing problems)  as a potential explanatory factor. 

I am also not entirely sure about the source of argument that "short-term economic shocks lead to more significant stress for a family than longer-term material hardships for which families may have had time to put coping strategies in place." (lines 496-497). Some studies would argue the opposite, using the "short-term resilience" argument.

Author Response

  1. In discussing the significant effect of housing hardship on cognitive outcomes (lines 488-502), I would strongly recommend considering access to schools / changing schools (due to housing problems)  as a potential explanatory factor. 

We agree that these are also important possible explanations and have added this to the discussion (lines 540-542).

  1. I am also not entirely sure about the source of argument that "short-term economic shocks lead to more significant stress for a family than longer-term material hardships for which families may have had time to put coping strategies in place." (lines 496-497). Some studies would argue the opposite, using the "short-term resilience" argument.

We appreciate that the reviewer highlighted this argument and have added this to the discussing, noting the need for additional research to understand the mechanisms and why short-term hardships appear more problematic for children’s behavior problems (lines 522-525).

Reviewer 2 Report

Excellent paper - well done to all involved. 

Author Response

Thank you.

Reviewer 3 Report

 Reviewer’s Comments on Child Poverty article

As a European with no experience or knowledge concerning the welfare systems of the USA, I have had some difficulty in understanding the context of this research, and have had to undertake some background reading.

One:  “poverty” in the article under review is defined in terms of Federal guidelines:

https://aspe.hhs.gov/topics/poverty-economic-mobility/poverty-guidelines/prior-hhs-poverty-guidelines-federal-register-references/2021-poverty-guidelines

According to these guidelines, at the current time a family of 2 adults and 4 children would be “poor” if their household income were $44660 pa, or less. This is puzzling,  since it is about twice the level of income defining “poverty” in many European countries for such a family. What does “poverty” actually mean in the Federal Government’s definition? Are the authors are actually considering families as “poor” whose income is 200% of the Federal guidelines? Again this seems barely credible as an indicator of “poverty”.

Two:  Why do “material poverty” and “income poverty” have so little overlap according to this Panel data? This is counter-intuitive, and requires a logical explanation, which is not offered in this article. Intuitively, my first guess would be measurement error in the statistical data. What other explanation could there be? Is there any ethnographic evidence on how “poor” families in the US organise their resources, and their lives, which explain this apparent paradox?

Three: Line 86 – 25% of US children are affected by “basic expenditure hardship”. How can this be in the world’s richest nation? A fuller explanation is required.

Four: Line 102 on “Medical Hardship”. Are the authors seriously suggesting that in the world’s richest nation, young children are denied medical care because of their parent’s poverty? This is hard to believe, and requires further explanation. Reference is made to CHIP, in which states provide some funding for basic medical care:

https://www.healthcare.gov/medicaid-chip/childrens-health-insurance-program/

My consultation of this document suggests that only very basic costs are covered, and these vary between States. 

No mention is made of children with severe or chronic health challenges from major disabling conditions. Did the Panel ask questions about this?

Five:  The PSID-CDS Panel study data refer to the years 1997 and 2002, so the data are 20 years out of date. Do any findings from the study have relevance today, or is this simply a historical study?  I note reference [1].

What is a Panel Study? The authors of the present article give no information on this. I also remain unsure of how the participants were enrolled, their ethnic profiles, and the areas they were recruited from. Was this a random sample? What was the setting of completion of questionnaires by parents? Is there any evidence on the reliability and validity of this information – surely a basic task in social and educational science? If these findings are to be taken seriously, we need answers to these questions.

Six:  Tables 3 to 6 present regression findings on the dependent variables, after several demographic variables have been (simultaneously or step-wise?) controlled for. But this information is not included in the Tables. I would be interested to know if, for example, whether ethnicity rather than material income etc explained more variance in the outcome measures. If for example, ethnicity is an important correlate of poverty, then perhaps it should be analysed as a primary rather than as a secondary variable. The Tables do not give us any clue as to whether this might be the case.

Seven: pages 544-545 references are given by authors names, not as numbers. I found it difficult to locate these authors in the reference list.

Eight: although some significant variance was explained in the outcome measures (behaviour and achievement of children) it is difficult to know whether these differences were problematic in terms of long-term child development. It could be that these differences, although significant, were actually without clinical or educational meaning. A fuller psychological perspective is required here.

Nine:  ‘Having to borrow money at end of the month’ is obscure. How often did this occur? Was the money borrowed from relatives, or from loan companies (at high interest rates)? Did food banks exist in the US for this purpose of helping poor families?

Ten:   Table 1, Variable “No money at end of the month”: the statistical  values  (12.83, SD 33.45) are exactly the same for the 1997 and 2002 cohorts. It is statistically possible for mean levels to be the same in both years, but it is statistically impossible for the standard deviation to be similar in both years. The mean values and SDs in this Table need to checked.

Eleven:  My statistical advice would be to divide the data for each cohort into income level quintiles, comparing a variety of variables across the five groups with ANOVA; and then conducting multiple regressions within each quintile group controlling on the  effect of each variable (poverty, ethnicity etc) as it enters the regression equation. This could tell  us much more about the meaning and strengths of relationships.

Twelve: I attach a recent study which may be of interest to the authors:

Bennett, D. L., Schlüter, D. K., Melis, G., Bywaters, P., Alexiou, A., Barr, B., ... & Taylor-Robinson, D. (2022). Child poverty and children entering care in England, 2015–20: a longitudinal ecological study at the local area level. The Lancet Public Health7(6), e496-e503.

Author Response

  1. “poverty” in the article under review is defined in terms of Federal guidelines:

https://aspe.hhs.gov/topics/poverty-economic-mobility/poverty-guidelines/prior-hhs-poverty-guidelines-federal-register-references/2021-poverty-guidelines

According to these guidelines, at the current time a family of 2 adults and 4 children would be “poor” if their household income were $44660 pa, or less. This is puzzling,  since it is about twice the level of income defining “poverty” in many European countries for such a family. What does “poverty” actually mean in the Federal Government’s definition? Are the authors are actually considering families as “poor” whose income is 200% of the Federal guidelines? Again this seems barely credible as an indicator of “poverty”.

We understood from this reviewer’s and the editors’ comments that there was a need to contextualize this US-focused study in the broader discourse about childhood poverty and hardship in wealthy countries. We had added a paragraph (lines 180-196) to the background section to provide readers with necessary context about poverty and poverty measurement in the United States relative to other wealthy countries and, in particular, European countries. We also added language to the discussion to specifically highlight this generalizability limitation of this study (lines 598-608).

Furthermore, we want to clarify that we consider families with an income that is between 100% and 250% of the FPL as near poor and not poor. The categories near poor and poor are commonly used in U.S. poverty research (see, for example, Duncan et al.’s seminal 2017 article “Moving Beyond Correlations in Assessing the Consequences of Poverty” among others). Finally, we appreciate the reviewer’s concern that $35,580 for a family of 6 or $44,660 for a family of 8 does not appear to be low income if directly compared to European incomes. However, costs of living are significantly higher in the U.S. with notably fewer government benefits than in European countries and, therefore, absolute poverty measures cannot be directly compared (see Smeeding et al.’s 2001 article “United States Poverty in a Cross-National Context”). Indeed, as we now highlight in the manuscript, the FPL tends to reflect a much lower income than is actually necessary to meet economic needs.

  1. Why do “material poverty” and “income poverty” have so little overlap according to this Panel data? This is counter-intuitive, and requires a logical explanation, which is not offered in this article. Intuitively, my first guess would be measurement error in the statistical data. What other explanation could there be? Is there any ethnographic evidence on how “poor” families in the US organise their resources, and their lives, which explain this apparent paradox?

The distinction between material hardship and income poverty has been widely documented using a wide range of data sources across countries (e.g., Fusco, Guio, & Marlier, 2011; Bradshaw & Finch, 2003; Iceland & Bauman, 2007; Lee & Lee, 2016; Notten & Mendelson, 2016 ; Schenck-Fontaine et al., 2020). The measurement and distinction of material hardship, as well as the recognition that material hardship does not directly reflect income poverty has its roots in European approaches to poverty measurement (Nolan & Whelan, 2010), which we now note in the manuscript on lines 190-196. There are many possible reasons for this apparent paradox, including that income poverty does not account for individual financial needs, assets, and economic behaviors. We now note this on lines 35-37.

  1. Line 86 – 25% of US children are affected by “basic expenditure hardship”. How can this be in the world’s richest nation? A fuller explanation is required.

That childhood poverty is so prevalent in the U.S. distresses the authors as well, but is a well-documented reality. Several excellent cross-country comparisons have been conducted on the topics of poverty and childhood poverty (see Rainwater & Smeeding’s “Poor Kids in a Rich Country” and Bradbury et al.’s “Too Many Children Left Behind”). It is precisely this unreasonably high rate of childhood poverty and the relatively weaker social safety net that makes the U.S. such an important context within which to study the effects of poverty and material hardship on children.

  1. Line 102 on “Medical Harship”. Are the authors seriously suggesting that in the world’s richest nation, young children are denied medical care because of their parent’s poverty? This is hard to believe, and requires further explanation. Reference is made to CHIP, in which states provide some funding for basic medical care:

https://www.healthcare.gov/medicaid-chip/childrens-health-insurance-program/

My consultation of this document suggests that only very basic costs are covered, and these vary between States. 

No mention is made of children with severe or chronic health challenges from major disabling conditions. Did the Panel ask questions about this?

Again, we share the author’s distress at the existence and prevalence of medical hardship among children in the U.S. Addressing the reviewer’s comment, we also now add that future work needs to be done to examine the co-occurrence of these material hardships among children with special health care needs and other populations vulnerable to higher levels of hardship (lines 563-566).

  1. The PSID-CDS Panel study data refer to the years 1997 and 2002, so the data are 20 years out of date. Do any findings from the study have relevance today, or is this simply a historical study?  I note reference [1].

What is a Panel Study? The authors of the present article give no information on this. I also remain unsure of how the participants were enrolled, their ethnic profiles, and the areas they were recruited from. Was this a random sample? What was the setting of completion of questionnaires by parents? Is there any evidence on the reliability and validity of this information – surely a basic task in social and educational science? If these findings are to be taken seriously, we need answers to these questions.

The reviewer rightly points out the need for additional information about the PSID-CDS in the manuscript. We have added additional information about the sample and the administration of the survey (lines 201-213). Racial and ethnic composition of the analytic sample is reported in Table 1. We also now point the reader to additional resources for further information about the PSID-CDS. Finally, we note we cite technical documentation for reliability and validity of measures used. If the editors prefer us to explicitly describe the measures’ reliability and validity in the manuscript, we are happy to do so.

  1. Tables 3 to 6 present regression findings on the dependent variables, after several demographic variables have been (simultaneously or step-wise?) controlled for. But this information is not included in the Tables. I would be interested to know if, for example, whether ethnicity rather than material income etc explained more variance in the outcome measures. If for example, ethnicity is an important correlate of poverty, then perhaps it should be analysed as a primary rather than as a secondary variable. The Tables do not give us any clue as to whether this might be the case.

The choice to focus on income poverty and material hardships as the primary variables of interest was intentional, as these are policy-relevant factors that can be feasibly addressed. Investigating race as a primary variable over and above the effects of income poverty and material hardship would only allow us to document existing race-based differences in children’s early outcomes. Not only are these race-based differences already documented extensively in existing studies (see, e.g., Burchinal et al., 2011), race-based differences are not themselves policy-relevant. Instead, such descriptive research of race-based differences presents only an important starting point that would require substantial additional investigation to understand the mechanisms that explain these differences. This would be, in essence, a different research question from that which is the focus of this particular study.

  1. pages 544-545 references are given by authors names, not as numbers. I found it difficult to locate these authors in the reference list.

We have fixed this.

  1. Although some significant variance was explained in the outcome measures (behaviour and achievement of children) it is difficult to know whether these differences were problematic in terms of long-term child development. It could be that these differences, although significant, were actually without clinical or educational meaning. A fuller psychological perspective is required here.

The reviewer is correct and we have added this to our discussion of the study’s limitations (lines 582-584).

  1. ‘Having to borrow money at end of the month’ is obscure. How often did this occur? Was the money borrowed from relatives, or from loan companies (at high interest rates)? Did food banks exist in the US for this purpose of helping poor families?

Unfortunately, the PSID-CDS data provide only information on the prevalence of borrowing from friends and family and its frequency. As such, it is true that families who have had to borrow money only once are categorized in the same way as families who had to borrow money multiple times. Additional information about the frequency of borrowing would be useful. However, we want to clarify that the question only asks about money borrowed from friends or family and does not include any formal loans or money borrowed from financial or other institutions. This is defined in line 261.

To answer the reviewer’s question about food banks, while these have existed and are used by families for many decades, there is evidence (see Schenck-Fontaine, Gassman-Pines, & Hill, 2017) that families combine multiple informal social safety nets to make ends meet and that food banks exist only to meet nutritional needs.

  1. Table 1, Variable “No money at end of the month”: the statistical  values  (12.83, SD 33.45) are exactly the same for the 1997 and 2002 cohorts. It is statistically possible for mean levels to be the same in both years, but it is statistically impossible for the standard deviation to be similar in both years. The mean values and SDs in this Table need to checked.

We have checked the data and corrected this typographical error.

  1. My statistical advice would be to divide the data for each cohort into income level quintiles, comparing a variety of variables across the five groups with ANOVA; and then conducting multiple regressions within each quintile group controlling on the  effect of each variable (poverty, ethnicity etc) as it enters the regression equation. This could tell  us much more about the meaning and strengths of relationships.

We appreciate the reviewer’s suggestion and have chosen to retain our original methodological approach. While mathematically, ANOVA and OLS regression are nearly identical, the information they provide is not. For the purpose of our research question, an OLS regression model was most appropriate, as it provides information about the effect sizes and not only information about how much variance is explained by the addition of different domains of material hardship.

  1. I attach a recent study which may be of interest to the authors:

Bennett, D. L., Schlüter, D. K., Melis, G., Bywaters, P., Alexiou, A., Barr, B., ... & Taylor-Robinson, D. (2022). Child poverty and children entering care in England, 2015–20: a longitudinal ecological study at the local area level. The Lancet Public Health7(6), e496-e503.

Thank you for drawing our attention to this very interesting new research. We will reference it in other work we are doing that is more closely related to the focus of that article.

Round 2

Reviewer 3 Report

I am satisfied that the authors have largely addressed my concerns. I would still like a paragraph to be added which offers a fuller design for a replicated study of this "historical" data set.

Author Response

We now include a paragraph (line 784-805) in the discussion that details the need to replicate these analyses using recent national data in the US. Unfortunately, as we note in the discussion, no such data currently exist, but their development needs to be prioritized to ensure the validity and relevance of research on childhood poverty and material hardship.